

# Prevalence of and related risk factors in oral mucosa diseases among residents in the Baoshan District of Shanghai, China

Shuyun Ge[1], Lin Liu[2,3], Qi Zhou[4], Binbin Lou[5], Zengtong Zhou[1], Jianing Lou[2,3,5] and Yuan Fan[2,3]

[1] Department of Oral Medicine, Shanghai Key Laboratory of Stomatology, Shanghai Ninth People's Hospital, Shanghai Jiao Tong University School of Medicine, Shanghai, China
[2] Department of Oral Medicine, Affiliated Hospital of Stomatology, Nanjing Medical University, Nanjing, China
[3] Jiangsu Key Laboratory of Oral Diseases, Nanjing Medical University, Nanjing, China
[4] Medical Association Office of Shanghai Baoshan District, Shanghai, China
[5] Department of Stomatology, Shanghai General Hospital of Nanjing Medical University, Shanghai, China

## ABSTRACT

**Background**. Oral mucosal diseases (OMDs) encompass a variety of different types of diseases. Our aim was to evaluate the prevalence and related risk factors of OMDs among residents in the Baoshan District of Shanghai, China, and provide a scientific basis for prevention and control strategies.

**Methods**. A sample of 653 residents aged 17 to 92 years from the Baoshan community was investigated in 2014. Each resident was surveyed by questionnaire to evaluate their oral mucosa and oral mucosa examinations were conducted. We followed up with 607 residents in 2018. All data were statistically analyzed using the SPSS 25.0 software package (Chicago, IL, USA) at the general population, gender and age levels. A $X^2$ test was used to compare rates of risk factors and logistic regression analysis was used to detect the correlation between disease and risk factors.

**Results**. The prevalence rate of OMDs was found to be 9.19%–9.56% (2014–2018). The most common OMDs were atrophic glossitis (1.84%), recurrent aphthous ulcer (RAU, 1.68%), burning mouth syndrome (BMS, 1.38%), oral lichen planus (OLP, 1.23%) and traumatic ulcers (1.23%). The prevalence of RAU and BMS in different age groups was significantly different. Tobacco and alcohol use and psychological factors in the OMDs group were higher than the no-OMDs group. Systemic diseases including diabetes mellitus (DM) was significantly relevant to OLP.

**Conclusion**. Age, tobacco and alcohol use, and psychological factor correlated strongly with the occurrence and development of OMDs, and they should be the focus of primary prevention. General epidemiological studies suggested that OLP was closely related to DM.

## INTRODUCTION

Oral mucosal diseases (OMDs) occur in the oral mucosa and most of them are associated with systemic diseases (*Porter, Mercadante & Fedele, 2017*). The various kinds of OMDs

Corresponding authors
Jianing Lou, 18939829711@163.com
Yuan Fan, fanyuan@njmu.edu.cn

include recurrent aphthous ulcer (RAU), burning mouth syndrome (BMS), oral lichen planus (OLP), oral leukoplakia (OLK), oral erythroplakia (OE), traumatic ulcers, and so on. Except for traumatic ulcers, the etiology of most OMDs is unknown. Some of these diseases have potential malignancies that seriously affect quality of life and are even fatal. Oral Potentially Malignant Disorders (OPMDs) refers to diseases that may cause oral cancer (*Wang et al., 2014*; *Dionne et al., 2015*). OLP is a common OPMDs, with a prevalence rate of 0.1%–4.0% (*Kurago, 2016*) and a cancerization rate of 0%–12% (*Van der Meij, Schepman & Van der Waal, 2003*). OLK is another common OPMDs, with a cancerization rate of 10–30% and an average time to cancer onset of 4.0–8.1 years (*Petti, 2003*; *Warnakulasuriya et al., 2011*). OE is the most cancer-prone OPMDs, with a cancerization rate as high as 50% (*Villa, Villa & Abati, 2011*).

Most of the causes of OPMDs are complex, and their pathogenesis and cancerous mechanism are unclear. Generally, there are no specific treatments and no effective chemical prophylaxis drugs for cancer (*Ribeiro et al., 2010*; *Van Monsjou et al., 2013*). OPMDs have a poor prognosis and the overall cancerization rate is close to 4.32% (*Wang et al., 2014*). Due to the prolonged course of the disorder, and the risk of cancer, patients often suffer physical and mental pain. Early diagnosis of OPMDs is of great significance for the prevention of oral malignant tumors (*McCullough, Prasad & Farah, 2010*; *Amagasa, Yamashiro & Uzawa, 2011*).

Shanghai, with a population of more than 24 million, was chosen as the location of the study because work stress, mental stress, environmental changes, dietary changes and accelerated life rhythms are thought to increase the prevalence of OMDs. Therefore, it is of great interest to understand the epidemiological characteristics of OMDs in this city and analyze the risk factors associated with those diseases. Early prevention and treatment of OMDs and early detection of high-risk groups of oral cancer will improve people's quality of life.

The prevalence of OMDs was studied in a cross-sectional study using general epidemiology. Our subjects were either selected from the general population in the Shanghai region or were oral outpatients. There is currently little available data on the OMDs in the Shanghai population. In the last 30 years, epidemiological investigations of OMDs have been mostly based on analysis of clinical data, or case-controlled studies (*Ikeda et al., 1995*; *Mumcu et al., 2005*; *Splieth et al., 2007*; *Pentenero et al., 2008*; *Mansour Ghanaei et al., 2013*; *Do et al., 2014*).

Little work has been done in the epidemiology of OMDs, especially descriptive epidemiology; most previous studies are investigations of specific age groups or special diseases (*Xu et al., 1981*; *Qi, 2008*). In order to make an accurate diagnosis of an OMD, good theoretical knowledge and sufficient clinical experience was required, especially considering the low prevalence of OMDs in the general population and large sample size in the survey.

In this article, we report on the epidemiological characteristics of OMDs in the Baoshan District of Shanghai, analyze the risk factors of the diseases to better understand prevalence and epidemic characteristics, and establish a scientific basis for the prevention and treatment of OMDs.

## MATERIALS AND METHODS

### Research objectives

We used multistage stratified random sampling and field surveys to investigate the population of the Baoshan District of Shanghai in 2014. Four neighborhood committees were selected by a random cluster method, and entire families were selected for oral examination according to house number. We calculated maximum sample size using the OMD prevalence rate of 14.93% (range 14.93–29.3% according to (*Xu et al., 1981*; *Cao et al., 1988*). The formula $[n = t2 * p/d(1 - P) 2, p = 14.93\%, d = 0.1 * p$, take $\alpha = 0.05, t = 1.96]$ yielded a theoretical figure of 2,189 people. Unfortunately, this study was only able to sample 653 people. We conducted follow up with patients 4 years later to assess the long-term effects of persistent risk factors on the course of the disease. The procedures were approved by the Ethical Committee of the Stomatological Hospital Affiliated to Nanjing Medical University (PJ2014-132) and the ethical committee of Shanghai First People's Hospital (2019KY063).

### Questionnaire design and oral mucosa examination

The survey was prepared according to the World Health Organization's (*WHO, 2013*) oral health assessment form guidelines. It was designed to evaluate the condition of the oral mucosa. The questionnaire includes demographics (name, gender, age, nationality, place of birth, length of residence in Shanghai, education level, marital status, occupation, etc.), smoking habits (current, former or never), drinking habits (current, former or never), systemic disease (diabetes, hypertension, coronary heart disease and cerebrovascular diseases), and mental status (stress, anxiety). For the purposes of the survey, we considered systemic disease to be identified if the resident was diagnosed by a doctor. Gestational diabetes was not included in this survey.

Oral mucosa examination followed the clinical diagnostic criteria proposed by the WHO (*Kramer et al., 1980*). Portable halogen lamps, disposable retractors, and mouth mirrors were used in this study. When performing oral mucosa examinations, the intraoral and perioral mucosa and soft tissue were comprehensively examined for each subject in the following systematic order: lip, corner of mouth, cheek, tongue, bottom of mouth, hard and soft palate, alveolar ridge and gingiva. When residents were clinically diagnosed with OPMDs, including OLP or OLK, we referred them to our hospital for laboratory tests and biopsies. Expert pathologists confirmed the final diagnosis. Informed consent was obtained prior to the initiation of the examination. Inspectors were trained to ensure that guidelines for observation were interpreted uniformly and that mucosal diseases were documented correctly. Calibration exercises were repeated every month. Ten inspectors participated in the study, six of those participated throughout the whole study.

### Statistical methods

All data were analyzed statistically using the SPSS 25.0 software package (Chicago, IL, USA). Two people were responsible for data entry and analysis. A $X^2$ test was used to compare the rates of risk factors. Logistic regression analysis was used to detect correlation between disease and risk factors. Results were considered significant if $p < 0.05$.

## RESULTS

### Survey results

A total of 653 permanent residents were surveyed in 2014, including 337 males and 316 females. We followed up with approximately 93% of those residents in the 2018 survey: 607 residents (311 men and 296 women). The remaining 7% were lost to follow up.

The education levels of the test subjects were mainly middle school (and below) level, mostly retired people. The average age of the respondents in 2014 was 66.05 years old (17–92), and the average age in 2018 was 67.38 years old (18–91).

### Prevalence of OMDs

In the 2014 survey, a total of 60 people (9.19%) had OMDs, which included 12 cases of atrophic glossitis (1.84%), 11 cases of RAU (1.68%), nine cases of burning mouth syndrome (BMS) (1.38%), eight cases of OLP (1.23%), eight cases of traumatic ulcer (1.23%), six cases of furrowed tongue (0.92%), three cases of geographic tongue (0.46%), one case of cheilitis (0.15%), one case of angular cheilitis (0.15%) and one case of leukoplakia (0.15%). The results showed that the OMDs were mostly localized at the back of tongue, buccal mucosa, labial mucosa and gingival (gums).

In 2018, 58 people (9.56%) had OMDs (46 people were lost to follow-up, which included 41 with no OMDs and five with OMDs). Compared to 2014, there were ten cases of decreased OMDs and nine cases of new OMDs as follows: 11 cases of atrophic glossitis (1.81%, one case lost, no new), nine cases of RAU (1.48%, two cases lost, no new), seven cases with furrowed tongue (1.15%, none lost, one new), nine cases of OLP (1.48%, none lost, one new case of DM), seven cases of traumatic ulcer (1.15%, none lost, eight cases of decrease, seven case new), one case of geographic tongue (0.16%, one lost, one case of decrease). One case each of cheilitis, angular cheilitis and leukoplakia (0.16%, none lost, no increase, no decrease).

### Distribution of diseases in males and females and in different age groups in 2014

The prevalence rate in females was higher than in males in cases of RAU, BMS and OLP ($p < 0.05$; Fig. 1). In addition, the overall prevalence of OMDs in females was also higher than in men ($p < 0.05$). The cases of RAU and BMS were statistically significant in different age groups. The prevalence of RAU in 15–29 year-olds and 30–39 year-olds was significantly higher than that in the older age groups (50–59, 60–69, 70–79) ($p < 0.01$), but there was no significant difference between the two groups ($p > 0.05$). We found BMS more frequently in people over 40 years old, prevalence increased with age, and was significantly higher in the 70–79 age group than in the 40–49 group ($p < 0.01$). OLP was detected in the population over 40 years old, and there was no significant difference among age groups (Fig. 2).

### Effect of living habits and mental state on OMDs in 2014

Respondents generally had a light diet. There were 107 (16.39%) smokers (people who had smoked for over three years) and 93 (14.24%) who were drinkers (millet wine and

**Distribution condition of various diseases in male and female**

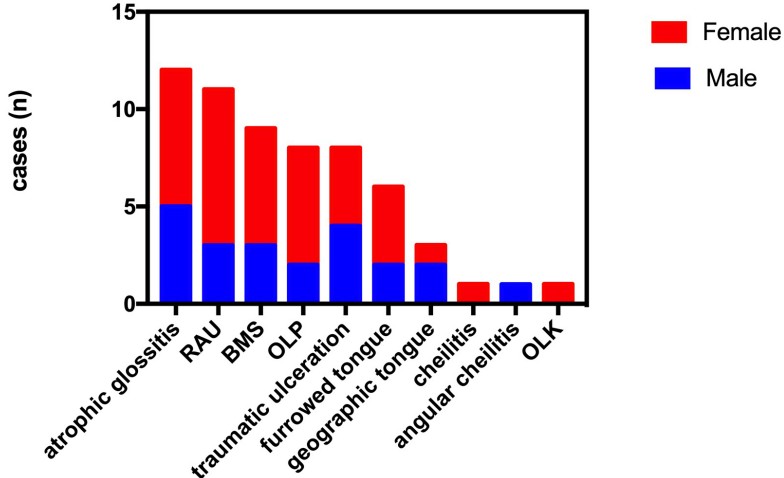

**Figure 1** Prevalence distribution of different types of OMDs among residents of different genders.

**Distribution condition of various diseases in various ages of resident**

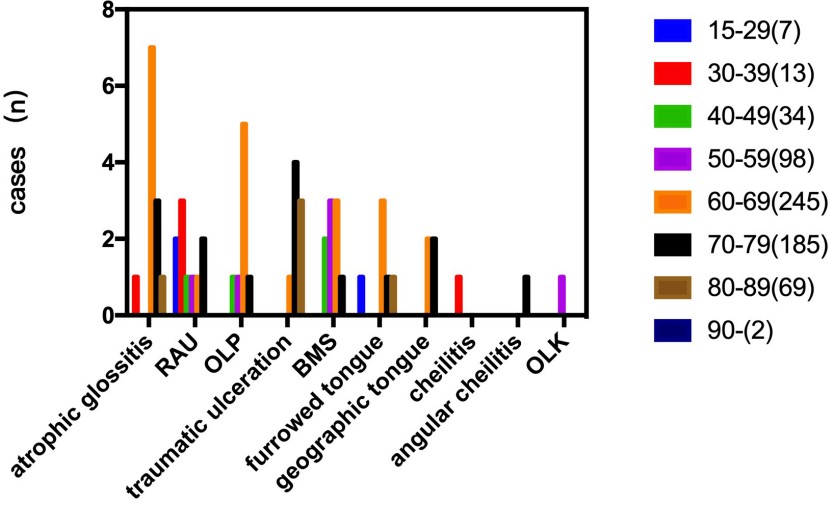

**Figure 2** Prevalence of different types of OMDs in residents of different age groups (disease prevalence trend of residents of different ages.

beer). The rate of smoking ($p = 0.01$) and alcohol consumption ($p = 0.00003$) in persons with OMDs was higher compared to those with no OMDs (Fig. 3). Our survey showed that 87 people were had mental anxiety and 24 people felt higher than normal stress in life. Those with OMDs were had a higher rate of anxiety compared to those with no OMDs

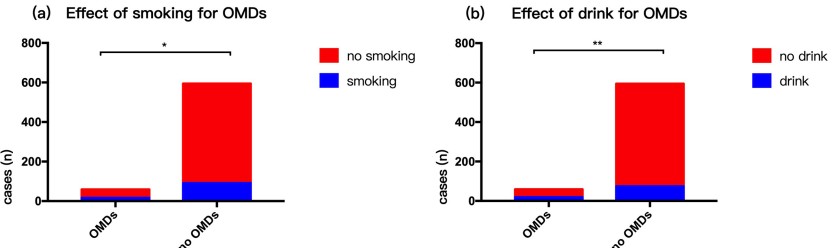

**Figure 3 Risk factors of oral mucosal disease in community residents—smoking and drinking habits.**
(A) Effect of smoking for OMDs. (B) Effect of drink for OMDs.

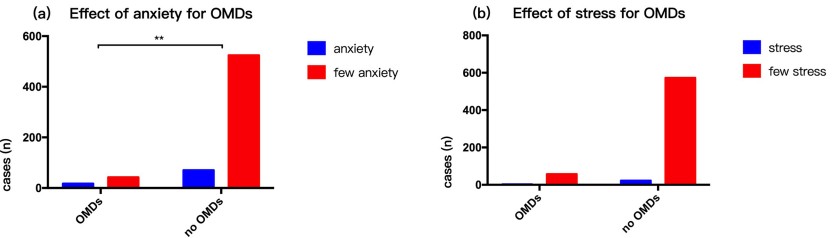

**Figure 4 Influence of mental state on OMDs of community residents.** (A) Effect of anxiety for OMDs.
(B) Effect of stress for OMDs.

($p = 0.0002$). However, there was no statistically significant difference in stress between persons with OMDs and no OMDs ($p = 0.903$) (Fig. 4). The sample group included 97.5% people in a stable family relationship, 28.1% with no friends and 36.8% having a little interpersonal communication. The vast majority (96.2%) of respondents had no adverse life experiences in past 12 months.

## Analysis of risk factors related to OMDs and OLP in 2014

This study found that there was no significant difference in the prevalence of OMDs between men and women in the population of Baoshan, Shanghai. It also found that the occurrence and prevalence of OMDs, particularly atrophic glossitis, RAU, BMS and OLP, were positively correlated with increasing age among residents.

The study also found that rates of smoking, drinking and mental anxiety in the OMD population were significantly higher than in the healthy population, but that there was no statistical difference. Systemic diseases, such as cardiovascular and cerebrovascular hypertension ($p = 0.025$) and metabolic diseases or diabetes ($p = 0.028$), were closely related to OMDs. Systemic diseases such as cardiovascular and cerebrovascular hypertension ($p = 0.025$), and metabolic diseases diabetes ($p = 0.028$) were closely related to OMDs. The investigation group theorized that the high number of elderly patients would have proportionally higher incidence of dental cavities and removable dentures and presumed that incidence of traumatic ulcers would also increase proportionally (Table 1).

**Table 1  Analysis of risk factors for OMDs.**

|  | Variable | Case | Control | Unadjusted OR[a] (OR 95% CI) | p value[a] | Unadjusted OR[b] (OR 95% CI) | p value[b] |
|---|---|---|---|---|---|---|---|
| age | ≥60 | 34(6.4) | 495(93.6) | 0.893(0.113–7.03) | 0.914 |  |  |
|  | 40–59 | 5(7.4) | 63(92.6) | 1.032(0.111–9.581) | 0,978 |  |  |
|  | <40 | 1(7.1) | 13(92.9) | 1 |  |  |  |
| smoking | yes | 5(7.0) | 66(93.0) | 1.107(0.417–2.934) | 0.839 |  |  |
|  | ex | 2(8.0) | 23(92.0) | 1.27(0.287–5.62) | 0.753 |  |  |
|  | no | 33(6.40) | 482(93.6) | 1 |  |  |  |
| drink | yes | 3(4.7) | 61(95.3) | 0.666(0.199–2.225) | 0.509 |  |  |
|  | ex | 0(0.0) | 9(100.0) | 0(0) | 0.999 |  |  |
|  | no | 37(6.9) | 501(93.1) | 1 |  |  |  |
| hypertension | yes | 14(4.7) | 287(95.3) | 0.533(0.273–1.041) | 0.066 | 0.45(0.224–0.906) | 0.025 |
|  | no | 26(8.4) | 284(91.6) | 1 |  |  |  |
| diabetes | yes | 10(10.5) | 85(89.5) | 1.906(0.899–4.042) | 0.093 | 2.41(1.098–5.294) | 0.028 |
| mellitus | no | 30(5.8) | 486(94.2) | 1 |  |  |  |
| coronary heart | yes | 4(5.2) | 73(94.8) | 0.758(0.262–2.192) | 0.609 |  |  |
| disease | no | 36(6.7) | 498(93.3) | 1 |  |  |  |
| cerebrovascular | yes | 3(8.8) | 31(91.2) | 1.412(0.412–4.837) | 0.582 |  |  |
| disease | no | 37(6.4) | 540(93.6) | 1 |  |  |  |

**Notes.**
[a] univariate logistic regression.
[b] multivariate logistic regression.

OLP was more common in the elderly age group (40–60 years) and occurred in more females than males. In this study, OLPs mostly occurred in the buccal region. The cause of OLP is complex and still unknown. The WHO defines OLP as a potentially malignant disease of the oral mucosa. Currently, there is no specific treatment for OLP in clinical practice. As a result, the course of the disease is prolonged and the condition often recurs. OLP may develop into cancer, which has a vital impact on patients' physical and mental health. The results of the general epidemiological investigation showed that there were eight community residents (1.23%) with OLP: two males and six females (male to female ratio 1:3). OLP was significantly associated with age, smoking and diabetes mellitus ($p < 0.01$) (Table 2).

## DISCUSSION

In this survey, we found that the prevalence rate of OMDs in the Shanghai region varied from 9.19% (2014) to 9.56% (2018). The most prevalent OMDs were incidence atrophic glossitis, RAU, BMS and OLP. Other surveys in Shanghai have found higher OMD prevalence rates: *Xu et al. (1981)* found a prevalence rate in 1978 of 14.93%; *Cao et al. (1988)* found a prevalence rate in the over 60s age group of 29.3%; and a large-scale epidemiological investigation conducted by *Feng et al. (2015)* among 11,054 people in 2012 found a prevalence rate of 10.8%.

**Table 2 Analysis of risk factors for OLP.**

| | Variable | Case | Control | Unadjusted OR[a] (OR 95% CI) | p value[a] | Unadjusted OR[b] (OR 95% CI) | p value[b] |
|---|---|---|---|---|---|---|---|
| age | ≥60 | 5(0.9) | 524(99.1) | 0.124(0.014–1.138) | 0.065 | 0.030(0.002–0.396) | 0.008 |
| | 40–59 | 3(4.4) | 65(95.6) | 0.6(0.058–6.230) | 0.669 | 0.286(0.022–3.717) | 0.339 |
| | <40 | 1(7.1) | 13(92.9) | 1 | | | |
| smoking | yes | 3(4.2) | 68(95.8) | 4.5(1.052–19.252) | 0.043 | 8.732(1.773–43.013) | 0.008 |
| | ex | 1(4) | 24(96) | 4.25(0.478–37.812) | 0.194 | 4.497(0.368–55.002) | 0.239 |
| | no | 5(1) | 510(99) | 1 | | | |
| drink | yes | 2(3.1) | 62(96.9) | 2.447(0.497–12.04) | 0.271 | | |
| | ex | 0(0) | 9(100) | 0(0) | 0 | | |
| | no | 7(1.3) | 531(98.7) | 1 | | | |
| hypertension | yes | 6(1.9) | 304(98.1) | 0.51(0.126–2.058) | 0.344 | | |
| | no | 3(1) | 298(99) | 1 | | | |
| diabetes | yes | 5(5.3) | 90(94.7) | 7.111(1.874–26.988) | 0.004 | 14.083(2.958–67.05) | 0.001 |
| mellitus | no | 4(0.8) | 512(99.2) | 1 | | 1 | |
| coronary heart | yes | 1(1.3) | 76(98.7) | 0.865(0.107–7.014) | 0.892 | | |
| disease | no | 8(1.5) | 526(98.5) | 1 | | | |
| cerebrovascular | yes | 1(2.9) | 33(97.1) | 2.155(0.262–17.747) | 0.475 | | |
| disease | no | 8(1.4) | 526(98.6) | 1 | | | |

**Notes.**
[a]univariate logistic regression.
[b]multivariate logistic regression.

Work, psychological stress, environmental changes, changes in diet structure and the accelerated pace of life are all thought to lead to an increased prevalence of the OMD condition.

In our study, OMDs were positively correlated with age, which supports the findings of *Cao et al. (1988)*. Therefore, preventive measures should be commensurate with the risk factor of increasing age. Atrophic glossitis occurred mainly in the elderly over 60 years of age in our study, which was consistent with its etiology: chronic anemia, lack of nicotinic acid, Sjogren's syndrome and Candida infection. Among people aged 30–39 years, there has been a significant increase in the prevalence of RAU which occurs in more female patients than male patients.

OLP and BMS also occurred in the older age group (40–60), and more women than men were ill.

In 2014, eight residents with traumatic ulcers were recommended treatment at the time of the first examination. During the 2018 follow-up, all eight patients dealt with trauma factors and their ulcers had disappeared. It indicated that community epidemiological surveys have, in addition to a diagnostic function, a preventative function that may contribute to the early treatment of residents, reduce cancer risk, and limit the number of untreated ulcers. The follow-up indicated that early prevention and control of mucosal diseases was of great significance. Seven new cases of traumatic ulcers found in the current investigation were treated and the patients were informed of the risks. Many of the interviewees were elderly people with high rates of dental cavities and periodontitis. The elderly lacked the

awareness to treat cavities and maintain dentures. As a result of the study, detection rates of traumatic ulcers increased, in particular, those occurring at the denture base covering the gums and palate areas. Glossy, traumatic ulcers were associated with residual roots and crowns. With continuous improvement in living standards, the rate of visits to medical centers to treat traumatic ulcers has gradually increased.

Our study showed that the prevalence of OMDs in patients who smoke, drink and experience anxiety was higher than in those who were healthy. Many studies have confirmed that tobacco and alcohol are closely related to OMDs and are risk factors for OMDs (*Shulman, Beach & Rivera-Hidalgo, 2004*; *Dundar & Ilhan Kal, 2007*; *Pentenero et al., 2008*; *Mohamed & Janakiram, 2014*). In past surveys, the prevalence of BMS was extremely low and rarely detected. However, the present survey found that the prevalence of BMS increased in middle-aged and older women in the perimenopausal and postmenopausal stages, and was accompanied by obvious changes in psychological and mental state. This might be due to the gradual acceleration of the pace of life in recent years, and subsequent increase in psychological problems. The causes of many diseases, especially chronic diseases, are no longer a simple biological factor, but also include many social, environmental and psychological factors. Alcohol, tobacco, stress and mental factors associated with the occurrence and development of OMDs should be the focus of primary prevention (*O'Sullivan, 2011*; *Mendes et al., 2012*).

It is noteworthy that the 4-year follow-up of community residents with OMDs suggested that systemic metabolic diseases, such as DM, were closely related to OPMDs, such as OLP. This result is consistent with the results of a large-scale oral health epidemiological investigation, in which we participated, in 2012 in Shanghai (*Feng et al., 2015*). The OMD group is comprised of many oral diseases and their prevention and control is difficult. At present, prevention and treatment of OMDs are still the main factors in the three-levels of disease control.

## CONCLUSION

Early prevention and control of mucosal disease is of great importance. It is necessary to better understand the prevalence of OMDs among Shanghai residents, and to carry out effective intervention activities through better health education, policy formulation and the creation of a supportive environment. Further, there is a need to reduce related risk factors, to promote the health of oral mucosa, and to improve the quality of life of the population. Finally, we consider it imperative to elevate the importance of primary prevention and treatment of OMDs.

## ACKNOWLEDGEMENTS

We thank all the physicians from the Department of Oral Mucosal Diseases, Ninth People's Hospital, Shanghai Jiao Tong University School of Medicine and Friendship Street Community Health Service Center of Shanghai Baoshan District for Participating in our epidemiological survey and performing the oral mucosal examination.

### Funding

This work was supported by the Science and Technology Development Fund Project of Baoshan, Shanghai (12E62), the National Natural Science Foundation of China- Youth Project (30700944), the Science Research Project of the Shanghai Health Bureau (2012092), the National Clinical Key Specialty Construction Project (2013-544), the National Institutes of Health (USA) Research Grant (R01 DK110273-01A1), the National Natural Science Foundation of China (81970941), and a project funded by the Priority Academic Program Development of Jiangsu Higher Education Institutions (PAPD 201887). The funders had no role in study design, data collection and analysis, decision to publish, or preparation of the manuscript.

### Grant Disclosures

The following grant information was disclosed by the authors:
Science and Technology Development Fund Project of Baoshan, Shanghai:  12E62.
National Natural Science Foundation of China- Youth Project: 30700944.
Science Research Project of the Shanghai Health Bureau: 2012092.
National Clinical Key Specialty Construction Project: 2013-544.
National Institutes of Health (USA) Research Grant:  R01 DK110273-01A1.
National Natural Science Foundation of China:  81970941.
Priority Academic Program Development of Jiangsu Higher Education Institutions: PAPD 201887.

### Competing Interests

The authors declare there are no competing interests.

### Author Contributions

- Shuyun Ge, Lin Liu and Jianing Lou conceived and designed the experiments, performed the experiments, analyzed the data, prepared figures and/or tables, authored or reviewed drafts of the paper, and approved the final draft.
- Qi Zhou performed the experiments, prepared figures and/or tables, and approved the final draft.
- Binbin Lou and Zengtong Zhou performed the experiments, authored or reviewed drafts of the paper, and approved the final draft.
- Yuan Fan conceived and designed the experiments, analyzed the data, prepared figures and/or tables, authored or reviewed drafts of the paper, and approved the final draft.

### Human Ethics

The following information was supplied relating to ethical approvals (i.e., approving body and any reference numbers):

Nanjing Medical University granted Ethical approval to carry out the study within its facilities (Ethical Application Ref: PJ2014-132).

### Field Study Permissions

The following information was supplied relating to field study approvals (i.e., approving body and any reference numbers):

Field experiments were approved by the ethical committee permission of Shanghai First People's Hospital (2019KY063).

### Data Availability

The raw measurements are available in the Supplementary Files.

### Supplemental Information

Supplemental information for this article can be found online at http://dx.doi.org/10.7717/peerj.8644#supplemental-information.

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
