# Peer review of "Prevalence of and related risk factors in oral mucosa diseases among residents in the Baoshan District of Shanghai, China"

_PeerJ, doi:10.7717/peerj.8644_

## Round 0.1 · original submission · Major Revisions

Your manuscript has been reviewed and requires modifications prior to making a decision. The comments of the reviewers are included at the bottom of this letter.

Reviewers indicated that the methods, results and discussion sections in the manuscript should be improved. For instance, Reviewer 1 asked how did you determine the sample size for this study and also found ethical committee permission was missed. Reviewer 3 also recommended extensive English editing. I agree with this evaluation and I would, therefore, request for the manuscript to be revised accordingly.

Reviewer 1 ·

Basic reporting

The article fails to meet the PeerJ standards
English is poor

Experimental design

The study is not well designed to demonstrate the aims and scope.

Validity of the findings

Provided data are not robust, not statistically supported.

Additional comments

The authors describes the prevalence of OMDs and related risk factors, in Shangai.
The study is not well designed and linguistic and conceptual limits do not raccomand the article publication in the present form.


INTRODUCTION

“… occur in the oral mucosa and soft tissues”
please specify the differences between oral mucosa and soft tissues.

“… and are related to general health”
not all the OMDs are related to general health. Traumatic lesions, for example, are not related to general health.

“…The etiology is unknown…” . Of what?

“…the course of disease is difficult to cure, easy to relapse, and is closely related to systemic diseases and physical and mental”.
Convulsionary phrase. Physical and mental what? Are you describing a specific disease or all the OMDs?

“OLP is the most common clinical OPMDs”
why have you specified “clinical”?

Please, cite the most common meta-analytic studies about the OLP rate of transformation in OSCC.

“Oral leukoplakia (OLK) is the most important OPMDs”:
why you defined OLK “important“ ?

Poor prognosis of OPMDs and the condition of multiple and recurrent cancers can cause great physical and mental pain to patients.
Poor prognosis of OPMD why? Please cite the references about physical and mental pain.

Early diagnosis of OPMDs with high risk of cancer is of great significance for the prevention of oral malignant tumors (McCullough et al., 2010; Amagasa et al., 2011).”
In my opinion, early diagnosis shlould be referred to all OPMDs, not only to OPMDs with high risk of cancer.

“In this study, the epidemiological characteristics of oral mucosal diseases..”
OMDs.

“…for the prevention and treatment of oral mucosal diseases.”
OMDs.

M&M
The sample size calculation is missing
Local ethical committee permission is missing in the text.
How the clinicians were calibrated?
Have you evaluated the inter-rater agreement? If yes, how?
“Our survey form was prepared according to the oral health assessment form of WHO (2013) (World Health Organization, 2013).”
Please cite a reference or the site address

Please describe how the patients were visited; how the lesions were scored or described, if biopsies were performed, were expert pathologists involved? Who confirmed the diagnosis?

RESULTS

“White collar person”
Please clarify
Please re-write results in a more appropriate way.

DISCUSSION
re-write according to results revisions.

·

Basic reporting

1. According to the authors, almost 7% of the participants were lost to follow-up; did these subjects differ in terms of social and demographic characteristics from those who were evaluated in 2018?
2. Figure 1, correct the word “make” with “male”. In the same Figure indicate significant differences between males and females.
3. In Figure 2, please present data not in lines but in columns.
4. In Figures 3 and 4 define the meaning of the asterisks.

Experimental design

How did the authors evaluate alcohol consumption and smoking? A definition should be provided in the methods section. The same stands for diagnosis of diabetes, hypertension, coronary heart disease and cerebrovascular disease.

Validity of the findings

The authors reported that they examined incidence of OMD during the 4-year period; however, no such information is provided and instead prevalence is reported.

Additional comments

This is an observational prospective study that examined the prevalence of oral mucosal diseases (OMDs) in residents in one district of Shanghai during the years 2014-2018. The authors examined 653 residents aged from 17 to 92 years old. They found that the prevalence rate of OMDs was 9.04-9.56%. The most common OMDs were atrophic glossitis (1.84%), recurrent aphthous ulcer (RAU, 1.68%), burning month syndrome (BMS, 1.38%), oral lichen planus (OLP, 1.23%), and traumatic ulcers (1.23%). Tobacco and alcohol use and psychological factor in OMDs group were higher than the no-OMDs group. System disease including diabetes mellitus (DM) was relevant to OLP. They conclude that age, tobacco and alcohol use, and psychological factor correlated with the occurrence and development of OMDs and that OLP was closely related to DM.

Comments:
1. According to the authors, almost 7% of the participants were lost to follow-up; did these subjects differ in terms of social and demographic characteristics from those who were evaluated in 2018?
2. How did the authors evaluate alcohol consumption and smoking? A definition should be provided in the methods section. The same stands for diagnosis of diabetes, hypertension, coronary heart disease and cerebrovascular disease.
3. Figure 1, correct the word “make” with “male”. In the same Figure indicate significant differences between males and females.
4. In Figure 2, please present data not in lines but in columns.
5. In Figures 3 and 4 define the meaning of the asterisks.
6. The authors reported that they examined incidence of OMD during the 4-year period; however, no such information is provided and instead prevalence is reported.

Reviewer 3 ·

Basic reporting

- English language and grammar use should be examined in the entire manuscript, including the tables

Experimental design

I have found one recent study reporting the prevalence of oral lesions in Shanghai, which is cited in the text and which included larger number of patients, but these patients were not followed-up after four years as the authors of this study did. This information from this article can be useful to the readers.
Please fill in, in the section Materials and Methods, how the participants were selected? How many inspectors examined the patients?

Validity of the findings

I have found one recent study reporting the prevalence of oral lesions in Shanghai, which is cited in the text and which included larger number of patients, but these patients were not followed-up after four years as the authors of this study did. This information from this article can be useful to the readers.

The first sentence in the "Conclusion section" should not begin with "Therefore", I suggest that the authors rephrase this sentence.

Additional comments

The study includes a large number of patients, and it is useful that the follow-up was made after four years to compare the data. After the suggested corrections and language correction, I think that the article can be published.

---

## Round 0.2 · Minor Revisions

Your manuscript has been re-reviewed and still requires modifications prior to making a decision.

The comments of Reviewer 3 are included at the bottom of this letter.

Reviewer 3 ·

Basic reporting

The authors have made a great improvement in English language and grammar. The sentences are now clear, easy to read and supported by data obtained from the research.

Experimental design

Research questions are well defined. The methods are described with sufficient details and information. My previous comments have been responded in the text.

Validity of the findings

The data are statistically sound and encouraged with the results from the literature. Conclusion is now well stated, reformulated and clear.

Additional comments

Dear authors, you have made significant improvement in structure and language. The limitation of the study is sample size, which you have pointed out in the text.The strength of your study is a four-year follow up of the patients, which is useful for comparing data.
I have noticed one small mistake in line 242, please correct this: "Our study showed that the prevalence of OMDs in patients who smoke, drink and experience anxiety was higher than in those who were healthy".
I recommend that, after this correction, the article should be accepted for publication.

---

## Round 0.3 · accepted · Accept

The authors addressed the reviewers' concerns and substantially improved the content of the MS. So, based on my own assessment as an academic editor, no further revisions are required and the MS can be accepted in its current form.